# A SWAT Evaluation of the Effects of Climate Change on Renewable Water Resources in Salt Lake Sub-Basin, Iran

**Sadegh Khalilian * and Negar Shahvari**

Department of Agricultural Economics, Faculty of Agriculture, Tarbiat Modares University, Tehran, Iran; negar_shahvari63@yahoo.com

**\*** Correspondence: sadegh.khalilian2019@gmail.com

**Abstract:** Future climate change is projected to have significant impacts on water resources availability in many parts of the world. This research evaluated climate change impacts on runoff, aquifer infiltration, renewable water resources, and drought intensity in Salt Lake sub-basin, Iran, by the Soil and Water assessment tool (SWAT) model and the Standardized Precipitation Index (SPI) under A1B, A2, and B1 climatic scenarios for 2011–2030, 2046–2065, and 2080–2099, using 1986–2016 as the reference period. The model was calibrated and validated by the SWAT-CUP software and SUFI-2 algorithm. Nash–Sutcliffe (NS) coefficients (0.58 and 0.49) and the determination coefficients ($R^2$) (0.65 and 0.50) were obtained for the calibration and validation periods, respectively. In order to study the climatic condition in the study basin, drought intensity was calculated. Then, drought intensity was predicted using the SPI index for the period 2011–2030. The results showed that runoff, infiltration, as well as renewable water resources will decrease under all climatic scenarios. Renewable water resources will be approximately reduced 100 Mm$^3$ by the year 2100. The future projections suggest a regional increase of 2 °C in temperature and a 20% decrease in precipitation in the sub-basin. In particular, drought intensity will be increased in the future. In 2015, this index was −1.31, and in 2016, the SPI index was lower than −2. These projection scenarios should be of interest to water resources managers in tropical regions.

**Keywords:** climate change; SWAT model; runoff; infiltration; renewable water; drought

## 1. Introduction

The management and planning of water resources are becoming more challenging as a consequence of the uncertainties of climate change [1]. The increase in temperature, variations in precipitation, and changes in the frequency of extreme events increase the probability of flood occurrences and change the total and seasonal water supply [2]. Iran is considered among the arid and semi-arid areas in the global climatic zonation [3]. Evidence from historic climatological data and forecasts of Iran's climatic conditions, as elsewhere worldwide, indicates climate change in recent decades with an ongoing trend in the future. Climatic alterations will seriously affect water resources, a large part of which, including surface waters, ground waters, and water-related constructions, will undergo unintended consequences. Simulations using hydrological, general circulation, or regional climate models are common methods for the study of the climate change impact on water resources. Abbaspour et al. (2009) assessed the impact of climate change on water resources in Iran. They found that the wet regions of the country will receive more rainfall, while the dry regions will receive less rainfall in scenarios A1B, A2, and B1 [4]. Ficklin et al. (2010) conducted a study on climate change effects upon basin runoff using the Soil and Water assessment tool (SWAT) model and the LARS-WG

software in order to investigate the effects of climate change on sediment load, nitrate, phosphate, and agricultural fertilizer residues in San Joaquin watershed, California, downscaled by LARS-WG for 2100. Through the Special Report on Emission Scenarios (SRES) regarding $CO_2$ emissions by agricultural practices, they found that the rising levels of this pollutant would result in a 23% drop in runoff and a 2 °C increase in temperature [5]. Faramarzi et al. (2013) conducted a research on climate change impact on freshwater availability in Africa. The results showed that for Africa as a whole, the mean total quantity of water resources is likely to increase. For individual subbasins and countries, variations will be substantial. Also, they showed that the number and frequency of dry days will increase in the future [6]. Devkota and Gyawali (2015) employed the SWAT model for hydrological simulation in order to investigate the climate change effect on the management and the hydrology of Kushi River basin in Nepal. Their results showed that climate change is not a large threat to available water in that area, though the projected flow was strongly dependent on the climate change implemented in the climate model [7]. Shrestha et al. (2016) applied the LARS-WG and SWAT models to study climate change-related runoff and sediment uncertainty in future periods of 2030 and 2060 under the Global Circulation Models (GCM) and reported an increase in sediment load and a decrease in runoff in the future [8]. Montaseri and Amiratae (2016) used 50 years of historical precipitation data at 12 stations in different parts of the planet to produce 1000 consecutive series of artificial rainfall and compared different methods of drought monitoring. Their results showed that the Standardized Precipitation Index (SPI) method is the most accurate and realistic indicator for drought analysis [9]. Leta et al. (2016) assessed the climate change impact on water balance components in Hawaii using the SWAT model. The predicted climate change scenarios showed that the decrease in rainfall during the wet season and the marginal increase in the dry season will be the main factors for the overall decrease in water balance components. Specifically, the groundwater flow may decrease by as much as 15% due to predicted rainfall and temperature changes by 2100 [10]. Alipour et al. (2017), in three central provinces of Iran, using 30-year precipitation data at 20 weather stations, showed that the SPI index is more flexible in drought analysis at monthly, seasonal, and annual time scales compared to other methods [11]. In another study, Kumar et al. (2017) investigated climate change impact on water resources in India. The results indicated that simulated annual discharge and percolation for 2020s will decrease by 2.9% and 0.8%, respectively [12]. Marcinkowski et al. (2017) conducted a study on climate change effects upon two Lowland Catchments in Poland. The hydrological response to climate warming and wetter conditions will include lower snowmelt, increased percolation, and baseflow and higher runoff [13]. The influences of climate changes on surface water resources in the Pangani basin were assessed by the Soil and Water assessment tools (SWAT) and Water Evaluation and Planning system (WEAP) models (Kishiwa et al., 2018). The results indicated a 10% rise in runoff and almost a 2 °C elevation in temperature during the 2050s compared to the baseline period. Irrigation is also predicted to undergo a severe shortage, necessitating current and future planning for water use [14]. Zhou et al. (2018) assessed the quantitative effects of climate change and anthropogenic activities on runoff fluctuations in Dongjiang River basin, China, and detected an elevated annual temperature and a reduced evaporation rate. The SWAT model also displayed an acceptable performance. Furthermore, climate change impacts (58%) have been slightly greater than those of anthropogenic activities (42%) in the whole basin [15]. Yin et al. (2018) employed the Climate Model Intercomparison Project 5 (CMIP5) and SWAT to investigate the climate change effects on Jinsha River flow. The results demonstrated a drop in runoff by 2–5% as a result of 1 °C temperature rise, with 0.5–0.8% decrease in precipitation and 1.31–1.87 °C elevation in temperature [16]. Using the SWAT model, the impact of climate change on rice yield was studied in Nanliujiang basin, China [17], suggesting a high ability of the SWAT model in simulating the studied basin. Rice yield increased from 1.4% to 10.6% under GFDL-ESM2M (Geophysical Fluid Dynamics Laboratory- Earth System Models) and IPSL-CM5A-LR (IPSL Earth system model for the 5[th] IPCC report) climate models, while HadGEM2-ES model (Hadley Global Environmental Multiscale Models) resulted in a dropped yield (Yang et al., 2018). In Awash basin, Ethiopia, Daba (2018) conducted a study on runoff sensitivity to temperature and precipitation and observed a high runoff

sensitivity to both variables, so that annual temperature rises of 1, 2, 3, 4, and 5 °C led to annual runoff reductions to −0.085, −0.88, −1.75, −2.55, and −3.30% [18]. In a study on the evaluation of climate change impact on Kan basin runoff (Hajimohammadi et al., 2018), runoff was simulated monthly by the SWAT hydrological model. The results indicated decreasing precipitation and rising temperatures in all selected stations. The developed climate scenarios finally demonstrated increasing and decreasing runoff levels in winter and other seasons, respectively. This present study evaluated climate change impact on runoff and drought in Salt Lake subbasins [19]. In another study, Montecelos-Zamora et al. (2018) assessed the potential impacts of climate change on water availability in the Cauto River basin. The future projections suggest a regional increase of 1.5 °C in annual temperature and a 38% decrease in annual precipitations in the subbasins. These changes translate to possible reductions in the annual streamflow of up to 61% with respect to the baseline period, whereas the aquifer recharge in the basin is expected to decrease up to 58%, with a consequent reduction of groundwater flow, especially during the boreal summer wet seasons [20]. Eini et al. (2018) evaluated runoff by the SWAT model and estimated drought intensity by the SPI index using Asfazari national database. Their results showed high accuracy of the Asfazari database in simulating runoff and drought severity [21].

Given different climate changes around the world, Iran cannot be excluded from these large-scale changes, the consequences of which are observed in many Iranian basins. As seen, very few studies have evaluated the effects of climate change on renewable water resources. The innovation aspect of this study is that we not only estimated the impacts of climate change on surface water, groundwater, and renewable resources, but also predicted drought intensity in future periods under different climate scenarios.

## 2. Materials and Methods

### 2.1. Case Study

This study was conducted in Salt Lake sub-basins, viz., Lavasanat, Damavand, and Varamin plain basins located at 35°0′0″ to 36°0′0″ N latitude and 51°0′0″ to 52°0′0″ E longitude (Figure 1).

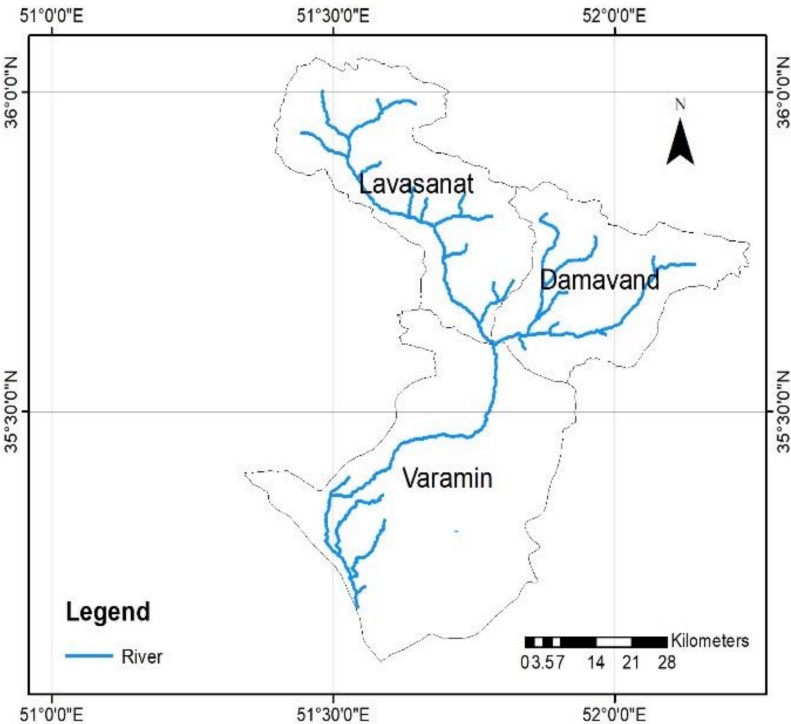

**Figure 1.** Study Basin.

Lavasanat basin (983 km$^2$) is a Salt Lake sub-basin located at the geographical coordinates of 35°45′ to 36°5′ N, 54°50′ to 51°20′ and 51°58′ E, with an average annual temperature of 13.5 °C. The annual precipitation reaches an average amount of 187 mm. Damavand basin (766 km$^2$) is another Salt Lake sub-basin situated at the geographical coordinates of 35°33′ to 35°52′ N and 51°47′ to 52°14′ E, with a mean annual temperature of 12.1 °C. The annual precipitation amounts to an average of 149 mm. Varamin basin (1720 km$^2$) is sited at the geographical coordinates of 35°7′ to 35°39′ N and 51°26′ to 51°55′ E, with a mean annual temperature of 16.9 °C. The annual precipitation averages an amount of 149 mm. The rivers Jajrod, Kandrod-Galandook, Damavand, and Ah are located in the study basin, the most important of which is Jajrod, wherein the Latian Dam is present upstream. The Jajrod river originates from the Alborz mountain ranges in the north of Tehran and runs southwest into the Latian Dam, then it joins the Damavand and Ah rivers ending into Salt Lake. The Latian Dam plays a major role in the hydrological cycle of the basin and also in downstream agricultural development. The dam with a watershed area of 69,800 km$^2$ and an average annual water flow of 350 mm$^3$ is located 35 km northeast of Tehran and 5 km from the Jajrod tiver.

## 2.2. SWAT Data

In this study, the SWAT hydrological model was developed using a 90 m DEM (Digital Elevation Model) layer, a land use layer of 2010, and a FAO v2 soil layer of 10 km (Figure 2A–C). In the first phase of modelling, the basin was divided into sub-basins based on topography and dividing line network in ArcGIS environment. Then, each sub-basin was divided into some hydrological response units (HRUs) according to land-use features, soil profile, and slope. The studied basin was divided into 68 sub-basins and 257 HRUs (Figure 2D).

Daily data of minimum and maximum temperatures and precipitation from three meteorological stations were introduced into the model to simulate the intended processes for the period 1986–2014 using SWAT software (Ver. 2012) as a program in ArcGIS 10.2 software. The model was then calibrated and validated using the monthly water yield from four hydrometric stations with SWAT-CUP software with SUFI-2 algorithm. The characteristics of the Latian hydrometric station are shown in Table 1.

**Table 1.** Characteristics of the Latian hydrometric station.

| Station | Station Code | Height | Longitude | Latitude |
|---------|--------------|--------|-----------|----------|
| Latian | 41119 | 1534 | 51°41′07″ | 34°46′32″ |

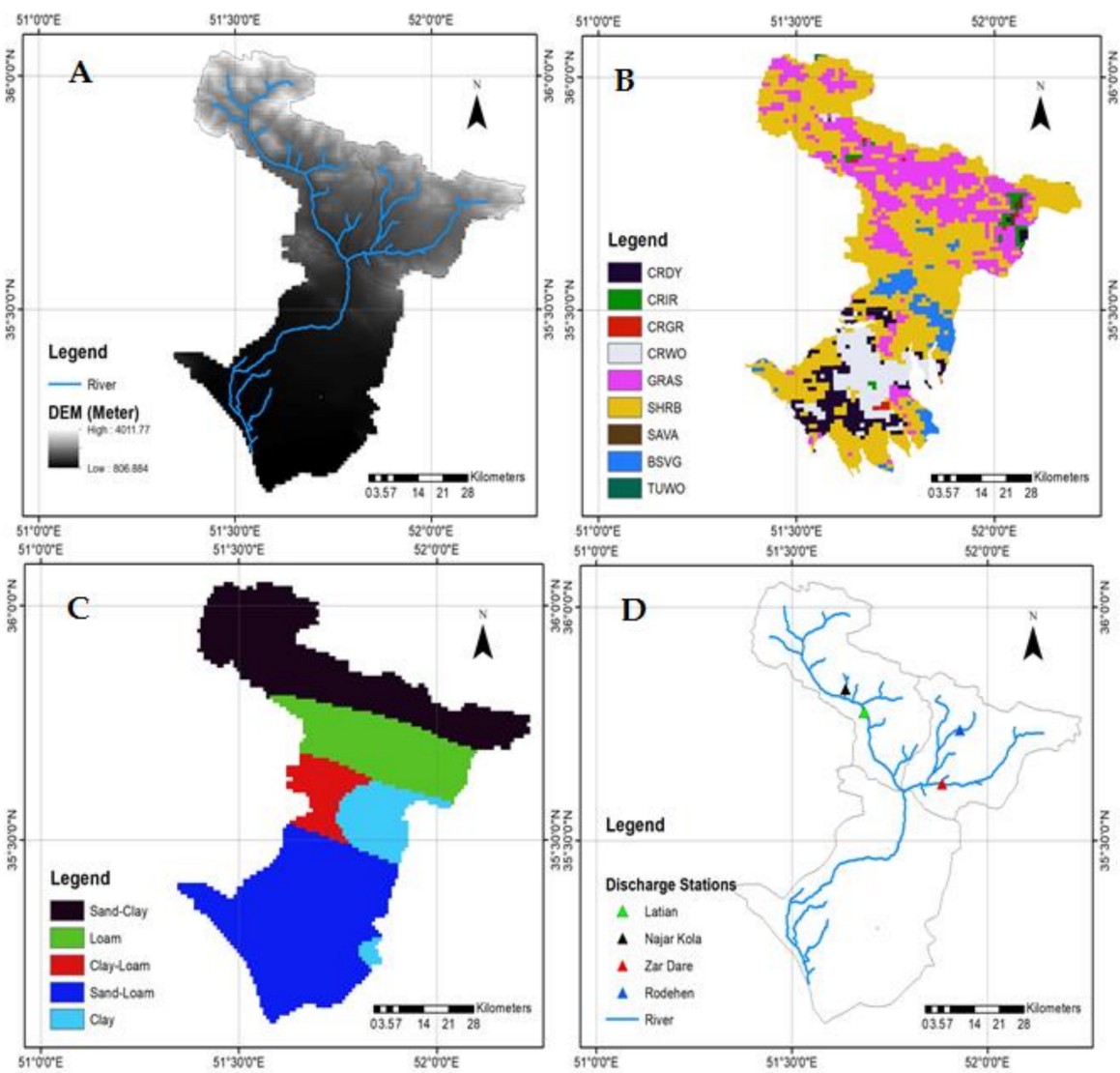

**Figure 2.** Digital Elevation Model (DEM) (**A**), land use (**B**), soil map (**C**), hydrometric stations (**D**).

## 2.3. Monthly Climatic Scenarios Generated Using AOGCM Models

Before using SWAT model, Atmosphere–Ocean Global Circulation Models (AOGCM) have been used to generate temperature and precipitation changes. Coupled three dimensional AOGCM are currently the most reliable tools for the generation of climate scenarios [22]. These models are based on physical relationships expressed by mathematical equations, which are solved in a three-dimensional network on the globe. To simulate global climate, the main components of the climate system (atmosphere, lithosphere, biosphere, and hydrosphere) are coupled in separate secondary models to form the AOGCM models [23]. The first series of emission scenarios, called IS92a–IS92f, was presented by the Intergovernmental Panel on Climate Change (IPCC) in 1992, which indicated increasing concentrations of greenhouse gases at a constant rate until 2100 [24]. A special report [25] offered an updated series of emission scenarios, called SRES. The report contains 40 scenarios for the world future divided into four main groups or scenario families (A1, A2, B1, and B2) based on economic-social advancements, accumulation of greenhouse gases, and suspended particles [26] (Table 2). LARS-WG is a model for the generation of random climate data applied for the estimation of daily precipitation production, radiation, and maximum and minimum daily temperatures in a station under present and future climate conditions [27–29]. The first version of LARS-WG was coined in Budapest, Hungary (1990), as a means of statistical downscaling. A random climate generator employs daily weather data recorded in a

station to compute a series of parameters for probability distributions of meteorological variables and their interrelationships.

**Table 2.** A summary of Special Report on Emission Scenarios (SRES) climate scenarios (A1, A2, B1, and B2) characteristics in 2100 compared to 1990.

| Scenario Feature | 1990 | B1 | A2 | A1 | B2 |
|---|---|---|---|---|---|
| World Population (Billion) | 5.252 | 7 | 15.1 | 7.1 | 10.4 |
| $CO_2$ Concentration in Earth's Atmosphere | 354 | 547 | 834 | 680 | 601 |
| Global Economic Growth index (GDP) | 21 | 328 | 243 | 550 | 235 |

*2.4. SWAT Simulation*

SWAT is a continuous model in basin scale designed to project the impacts of various management strategies on water levels, sediment, and chemical-agricultural substances in vast and complex basins with different soil, land use, management, and morphological conditions in the long term. It is a physical–distributional model for the assessment of soil and water issues. Instead of associating regression equations to describe input–output interrelationships of variables, the model uses data related to weather, soil profile, topography, vegetation, management strategies, and land use in the basin. The model indirectly simulates physical processes associated with water flow, sediments, plant growth, nutrient cycle, and so, on using input parameters. Computationally, it is a highly efficient model that simulates large and complex basins with the least time expenditure, enabling the user to address the long-term impacts of such factors as sediment, pollution, and erosion. This model is based on a comprehensive and robust database for the above different components. This database determines definitions and limits of parameters related to the above various components, then the locations of synoptic and climatological stations and their data, as well as introduces into the model different management strategies of water resources and current agriculture in the study area. The model inputs are generally categorized into point and spatial data.

The SWAT model uses the water yield equation (Equation (1)) to simulate the hydrological cycle. The simulated hydrological processes include evapotranspiration, runoff, snowmelt, surface seepage, deep seepage, groundwater flow, and subsurface flows.

In this research, HRUs were first defined and then divided by introducing the above maps. Thereafter, parameters related to each main component, namely, vegetation (DAT), edaphic (Sol), groundwater (GW), management (Mgt), and riverine (Rte) were introduced into the model.

$$SW_t = SW_0 + \sum_{i=1}^{t} (R_{day} - Q_{surf} - E_a - W_{seep} - Q_{gw})_i \tag{1}$$

where $SW_t$ is the final soil water content on day t, $SW_0$ is the initial soil water content, t is time (days), $R_{day}$ is the precipitation rate on day t, $Q_{surf}$ is the runoff level on day t, $E_a$ is the evapotranspiration on day i, $W_{seep}$ is the water seepage from root zone on day i, and $Q_{gw}$ is the returned flow on day i.

*2.5. Calibration Analysis*

Because there are many parameters in the SWAT model and because of the concurrent simulation of many hydrological and agricultural variables in this model, a new model, called SWAT-CUP, was developed, its sensitivity was analyzed, and calibration was performed. There are two analyses of local sensitivity, either related to time or global sensitivity. In local sensitivity analysis, one input is changed within predefined limits while keeping other inputs constant, then model output changes are examined depending on changes in each parameter. The parameter with a higher absolute value of t-stat and a p-value close to zero will have a greater effect on the variable. The model was calibrated by the coefficient of determination ($R^2$) and the NS coefficient.

*2.6. SPI Index*

The standard precipitation index (SPI) was first introduced in 1993 by McKee et al. This index is calculated for each region on the basis of the record of its long-term rainfall. Rainfall is the only data required for drought testing using the SPI index [30]. At first, the appropriate statistical distribution is fitted with the long-run rainfall data. Then, the cumulative distribution function is converted to normal distribution using equal probabilities (Equation (2)).

$$\text{SPI} = \frac{X_i - X^-}{S_n} \tag{2}$$

$X^-$ is the average rainfall per month, $S_n$ is the standard deviation on time scale, and $X_i$ is the rainfall per month.

Because of the normality of the SPI index, dry and wet climates can also be monitored using the SPI Index. Table 3 lists the drought classification based on the SPI index.

**Table 3.** Drought classification.

| SPI Values | |
| --- | --- |
| +2.0 and more | extremely wet |
| 1.5 to 1.99 | very wet |
| 1.0 to 1.49 | moderately wet |
| −0.99 to 0.99 | near normal |
| −1.0 to −1.49 | moderately dry |
| −1.5 to −1.99 | severely dry |
| −2 and less | extremely dry |

## 3. Results and Discussion

*3.1. Sensitivity Analysis of the Model Parameters*

The period 1998–2014 was selected after statistical analysis of the data from the climatological and hydrometric stations in the studied basin and also considering the research objective and the model's need of inputs with continuous time paces and simultaneous time series. Of this statistical period, the years 1998–2011 were considered for calibration, and three last years (2011–2014) were used to validate the model. Then, SWAT model calibration and validation were performed through the preparation of monthly time series of the measured data using the SWAT-CUP software and the SUFI-2 algorithm. A total of 103 sensitive parameters were calibrated and validated to simulate runoff in the model.

The parameters with relatively greater impacts on the streamflow are shown in Table 4. The letters V and R are the codes that determine the type of changes applied to a parameter, that is, v denotes the replacement of a parameter value with a new value of a new parameter, and r is the parameter value multiplied by (1 + given value), replacing the parameter. Table 3 shows the effects of some important parameters involved in the simulation of stream flow in the sub-basins together with *p*-values and t-stats. The parameter with a relatively higher t-stat absolute value and a *p*-value close to zero has had a greater impact on the stream flow.

**Table 4.** Sensitivity analysis and determination of the effective parameters of the model; V: replacement of a parameter value with a new value of a new parameter; R: parameter value multiplied by (1 + given value)

| Parameter Name | Parameter Definition | t-Stat | *p*-Value |
|---|---|---|---|
| R__CN2.mgt | Scs runoff curve number | −23.99 | 0 |
| V__PLAPS.sub | Precipitation lapse rate(mm/km) | 20.84 | 0 |
| R__SOL_BD( . . . ).sol | Moist bulk density (mg/m$^3$) | 3.92 | 0 |
| V__CH_N2.rte | Manning's n value for main channel | −3.8 | 0 |
| V__ALPHA_BF.gw | Base flow alpha factor (days) | 1.92 | 0.06 |
| V__HRU_SLP.hru | Average slope steepness (m/m) | 1.92 | 0.06 |
| V__LAT_TIME.hru | Lateral flow travel time (days) | −1.88 | 0.06 |
| V__SLSUBBSN.hru | Average slope length | −1.62 | 0.11 |
| V__RCHRG_DP.gw | Deep aquifer percolation fraction | 1.57 | 0.12 |

*3.2. Calibration and Validation of the SWAT Model*

After the sensitivity analysis, the model was calibrated and validated using monthly statistics from Latian hydrometric stations. Model simulations were evaluated by the coefficient of determination ($R^2$) and the NS coefficient (Table 5). As shown in Figure 3, despite the vastness of the basin, the final calibration values indicate the ability of the SWAT model to appropriately simulate the basin.

**Table 5.** Calibration and validation results for each hydrometric station in the period 1987–2014. $R^2$: coefficient of determination; NS: Nash–Sutcliffe coeffcient

| Station | River | Station Code | $R^2$ | NS | $R^2$ | NS |
|---|---|---|---|---|---|---|
| | | | Calibration | | Validation | |
| Latian | Jajrod | 41119 | 0.65 | 0.58 | 0.5 | 0.49 |

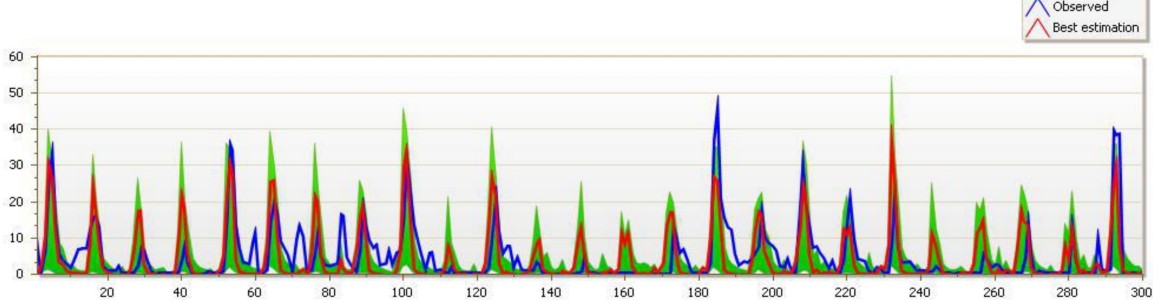

**Figure 3.** Observed and simulated time series graph with 95% probability band (Latian hydrometric station).

*3.3. Annual Runoff Simulation*

As SWAT model needs both parameters at the same time, HadCM3 was introduced as the selected model, the data of which were applied in all phases. The scenarios were selected on the basis of drawing future climate conditions. Therefore, A1B indicated a temperate weather, A2 presented the most critical condition in parameter estimation, and B1 provided more optimistic consequences than the other two scenarios following climate change. Following LARS-assisted downscaling of climate data for future periods under the above scenarios and HadCM3 model, a 30-year time series of the data for all three future periods was prepared to be introduced into the SWAT model. Afterwards, the SWAT output with applied coefficients was analyzed by SWAT-CUP, and the annual runoff level was simulated based on the baseline period duration for future periods under A1B, A2, and B1 scenarios (Figure 4). The results of the LARS-WG model indicated that the mean annual temperature will increase 2 °C, and the mean annual precipitation will decrease 20% in the future. Moreover, as shown in Table 6, a decrease of runoff will

lead to a reduction in surface water. The greater reduction will occur in the period of 2080–2099 under the A2 scenario. A 20% reduction in rainfall will lead to reduction in runoff. Reducing surface water will not only decrease the water availability resources, but also have a negative impact on agriculture, because all surface water in this basin will be consumed by the agricultural sector.

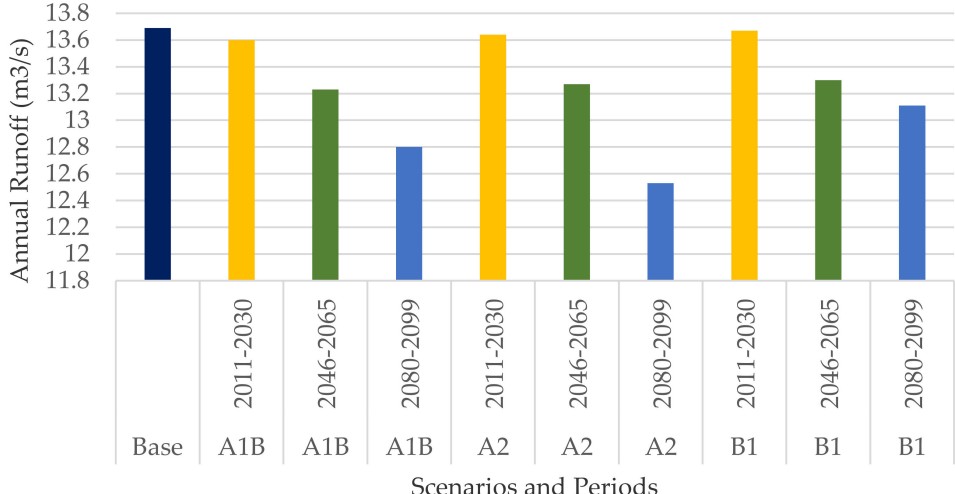

**Figure 4.** Simulation of runoff under the examined climate scenarios.

**Table 6.** Simulation of runoff and surface water under the examined climate scenarios.

| Scenario | Period | Annual Runoff ($m^3$/s) | Surface Water ($Mm^3$) |
|---|---|---|---|
| Base | | 13.69 | 431.3 |
| A1B | 2011–2030 | 13.6 | 428.8 |
| | 2046–2065 | 13.23 | 416.2 |
| | 2080–2099 | 12.8 | 403.6 |
| A2 | 2011–2030 | 13.64 | 428.8 |
| | 2046–2065 | 13.27 | 418.4 |
| | 2080–2099 | 12.53 | 395.1 |
| B1 | 2011–2030 | 13.67 | 430.2 |
| | 2046–2065 | 13.3 | 419.4 |
| | 2080–2099 | 13.11 | 413.4 |

*3.4. Aquifer Percolation Simulation*

After determining the model's ability to simulate runoff, the aquifer infiltration was calculated for the baseline and future periods. During the 30-year period, the groundwater level in the study area dropped 37.50 m, with an average annual rate of 1.47 m. As shown in Table 7, the groundwater recharge will be decreased under all climate scenarios because of climate changes. On the other hand, water discharge will be constant (405 $Mm^3$). The lowest amount of aquifer recharge will occur in the 2080–2099 period under the A2 emission scenario, whose recharge is 281 $Mm^3$ (Figure 5).

Dropdown of groundwater in the study basin will have adverse consequences such as decreasing the water resources, increasing the wells depth, and drying some fields and gardens, which are in conflict with the goals of sustainable development of agriculture.

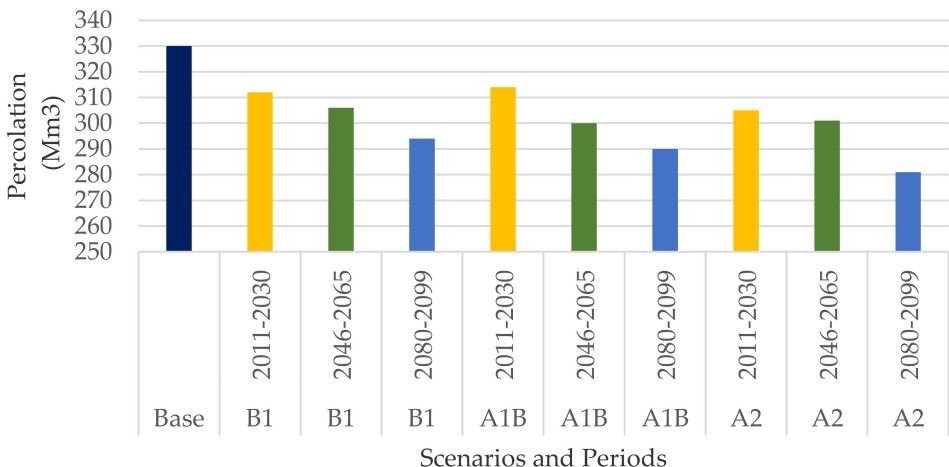

**Figure 5.** Simulation of aquifer percolation under the examined climate scenarios.

**Table 7.** Simulation of aquifer percolation under the examines climate scenarios.

| Scenario | Period | Groundwater Recharge (Mm$^3$) | Discharge (Mm$^3$) |
|---|---|---|---|
| Base | | 330 | 405 |
| B1 | 2011–2030 | 312 | 405 |
| | 2046–2065 | 306 | 405 |
| | 2080–2099 | 294 | 405 |
| A1B | 2011–2030 | 314 | 405 |
| | 2046–2065 | 300 | 405 |
| | 2080–2099 | 290 | 405 |
| A2 | 2011–2030 | 305 | 405 |
| | 2046–2065 | 301 | 405 |
| | 2080–2099 | 281 | 405 |

*3.5. Renewable Water Resources Simulation*

Renewable water resources include surface water and aquifer infiltration. The results showed that, because of the decreasing in runoff and recharge, the renewable water will be decreased in the future periods considered. Unfortunately, it will decrease more than 100 Mm$^3$ during 80 years under the A2 scenario, as shown in Table 8. Moreover, the greatest decline in renewable water in the near future will happen under the A1B scenario (Figure 6). Reducing renewable water resources will have undesirable impacts on water availability.

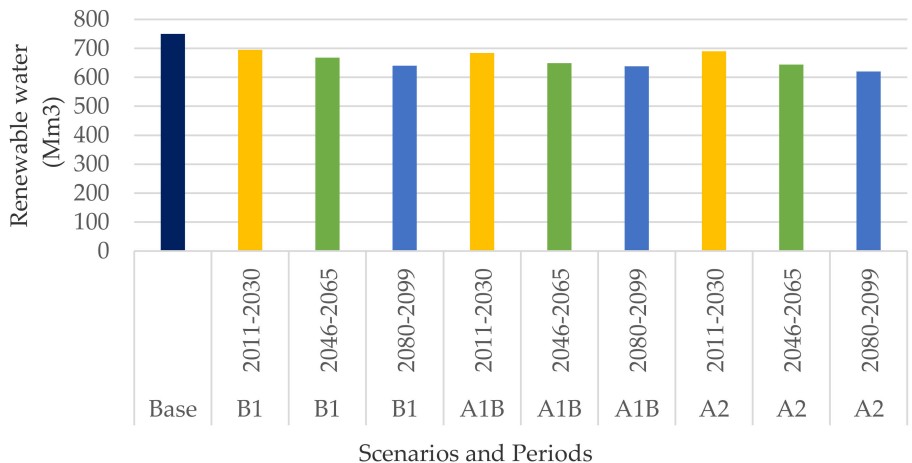

**Figure 6.** Simulation of renewable water under the examined climate scenarios.

**Table 8.** Simulation of renewable water under the examined climate scenarios.

| Scenario | Period | Renewable Water (Mm$^3$) |
|---|---|---|
| Base | | 750 |
| B1 | 2011–2030 | 695 |
| | 2046–2065 | 668 |
| | 2080–2099 | 640 |
| A1B | 2011–2030 | 684 |
| | 2046–2065 | 649 |
| | 2080–2099 | 638 |
| A2 | 2011–2030 | 690 |
| | 2046–2065 | 644 |
| | 2080–2099 | 620 |

*3.6. Drought Severity*

One of the negative impacts of climate change is the occurrence of drought. In order to investigate the drought phenomenon and its severity in future periods, the 12-month SPI index was estimated for baseline setting (1986–2016). Years when the SPI number is less than zero are years of drought, whereas those when this number is higher than one are mild periods. As displayed in Figure 7, the largest mild period lasted 5 years. Since 2008, relatively severe droughts occurred in the plain of Varamin. Since 2013 onwards, the drought became very severe during, and the SPI reached the value of −1. In 2015, this figure was −1.31, and in 2016, the SPI was lower than −2. The greatest drought hd happened in the base year of 1997, with an SPI of −3. On the other hand, the best mild period was in 1996, with an index over +2.

After the SPI was calculated for the period 1986–2016, the drought intensity was estimated for the period 2011–2030 under A1B, A2, and B1 scenarios. As shown in Figure 8, in the years 2023 and 2027, there will be severe drought in the study area. The most severe drought will happen in 2027 under the A2 scenario. The drought will have negative impacts on water resources and agriculture of the region. The assessment of the number of dry years and the frequency of their occurrences suggests an increase in the drought events and their duration in the future.

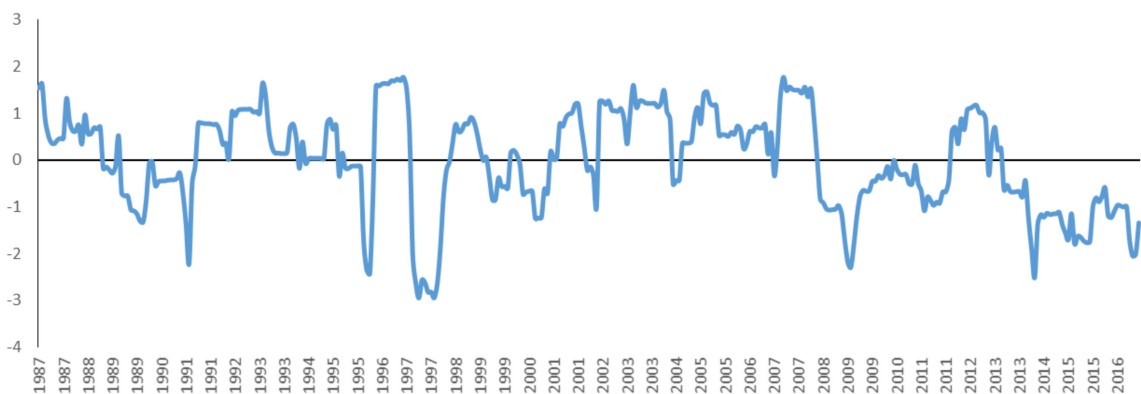

**Figure 7.** Standard precipitation index (SPI) chart of Varamin plain (1987–2016).

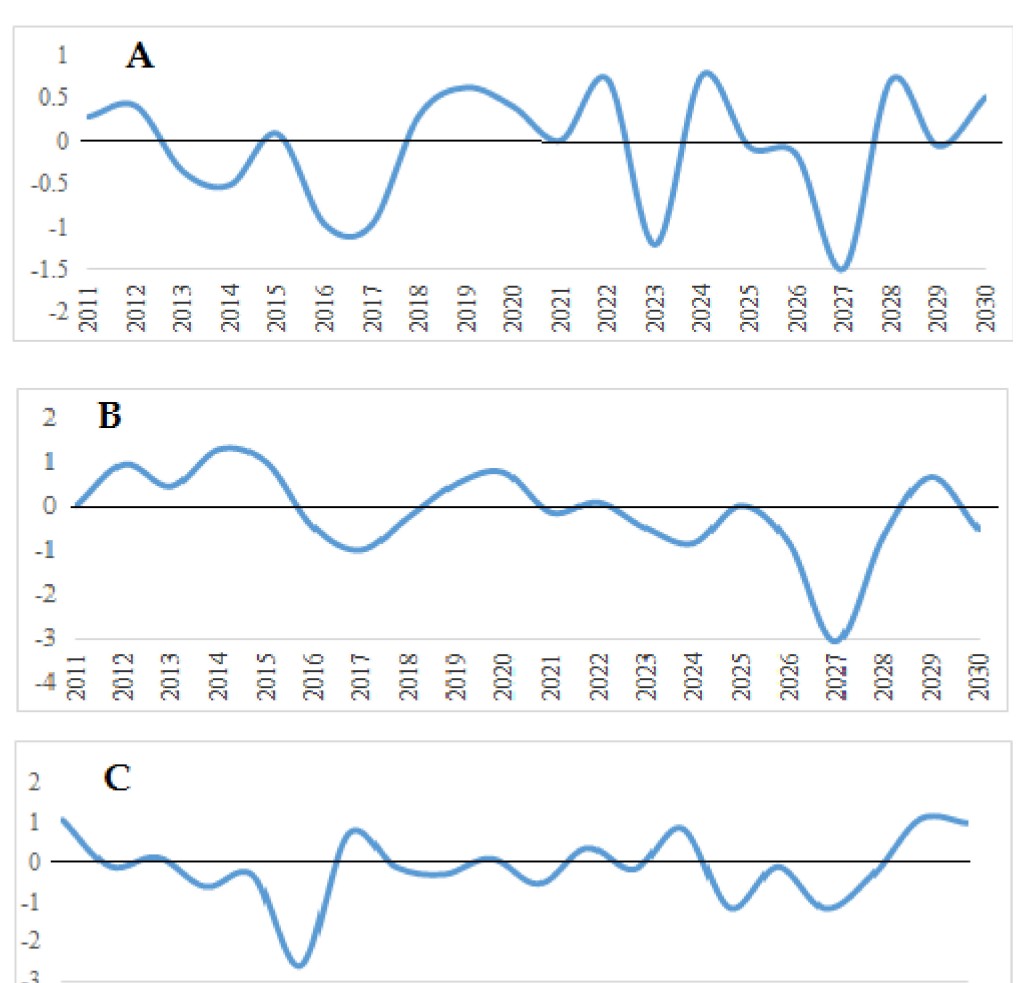

**Figure 8.** The SPI index for the period 2011–2030 under A1B (**A**), A2 (**B**), B1 (**C**) scenarios.

## 4. Conclusions

This study examined the climate change impact on the surveyed basin drought, runoff, percolation, and renewable water resources during the periods 2011–2030, 2046–2065, and 2080–2099, using a hydrological (SWAT) model. The results showed that severe droughts will be observed in the study basin. Specifically, in the year 2027, the drought intensity will be extremely severe under the A2

scenario, with and SPI of $-3$. On the basis of these future projections, there will be an increase of $2\,^\circ\text{C}$ in the mean annual temperature and a 20% decrease in the mean annual precipitation. The predicted climate change scenarios showed that the decrease in rainfall and increase in temperature will be the main factors for the decrease in water availability. According to the A2 scenario, the surface water resources in the study area will decrease 10%. Although the discharge from wells will remain constant, the SWAT model also suggested a reduction of the aquifer recharge of 16% and of renewable water of 20% due to climate change, which may have serious implications on groundwater availability in the study area. Also, drought will have inevitable impacts on water resources. This poses additional challenge to agriculture and food production. If no measures are taken, the water status of the area will be critical. These projection scenarios should be of interest to water resources managers in tropical regions. Moreover, the knowledge of a possible reduction of runoff, aquifer recharge, and renewable water is strategic for adaptation actions in the study area.

**Author Contributions:** All authors contributed to all stages of the research.

**Funding:** This research received no external funding.

**Conflicts of Interest:** The authors declare no conflict of interest.

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
