# Peer review of "A SWAT Evaluation of the Effects of Climate Change on Renewable Water Resources in Salt Lake Sub-Basin, Iran"

_agriengineering, doi:10.3390/agriengineering1010004_

Round 1
Reviewer 1 Report
Line 12: Be consistent in your text, consider rephrasing as " the determination coefficients (R2)...
Line 16: no need to put brackets around MM
Line 21: Rephrasing " ... among the arid and ...."
Line 23: Rephrasing " ... indicating that the incidence ..."
Line 25: Replace "remarkable" with "unintended"
Line 27: Replace "among" with "common"
Line 31: Rephrase as "Through the Special ... "
Line 35: Replace "hydrological water resources" with just "the hydrology"
Lines 40, 41: Rephrasing as "... used 50 years of historical precipitation data ..."
Line 46: Delete the sentense "It showed itself."
Line 65: use lower case "r" for runoff.
Line 66: Rephrase "... have indicated that decreasing precipitation ..."
Line 67: Rephrase as "Salt Lake sub-basins" I assume Salt Lake is an actual name of the lake?
Line 72: add (SPI) after "The standard precipitation index"
Line 72: "et al.", not "at al."
Line 109: Rephrase as "... the basin was divided into sub-basins based on ..."
Lines 165 - 168: After introducing the water yield equation, add the word "where" before explaining the variables. Also, do not sure ":" here, write out each variable in sentence.
Section 3.1: This section need complete rephrasing and expanding. It's very confusing and the emission scenarios results weren't broken down in detail for explanation.
Line 193: Rephrasing as "... was selected after statistical analyses..."
Line 202: Rephrasing as " After sensitivity analysis"
Sections 3.3 - 3.5: These sections need expanding and much more elaborations. Need a much detailed interpretation of result and discussion. For instance, what are the driving forces and mechanisms to yield these results? Also, discuss and interpret each emission scenario result in much greater detail.
Line 240: Delete "Because"
Line 240: Use lower case "a" for aquifer
Line 241: Rephrase as "climate change", no need to use "changes"
Section 4: Again, what emission scenario is the conclusion base of? If it's all of them, you need to indicate here.
Author Response
Dear First Referee,
Thank you for your time and recommendations. I have addressed your comments as follow:
1. Line 10: “The determination coefficients ( was rephrased.
2. Line 15: “Bracket” removed.
3. Line 26: “The” was added.
4. Line 28: “Indicating that” was rephrased.
5. Line 31: The word “Unintended” was replaced.
6. Line 32: The word “Common” was replaced.
7. Line 39: “The” was added.
8. Line 47: “The hydrology” was replaced with “hydrological water resources”.
9. Line 53: “Used 50 years of historical precipitation data” was rephrased.
10. “It showed itself” removed.
11. Line 87: “r” replaced “R”.
12. Line 88: “have indicated that decreasing precipitation” rephrased.
13. Line 91: “Subbasins” replaced subbasin.
I assume Salt Lake is an actual name of the lake?
Response 13: That is right. Salt is the name of one of the lakes in Iran, which has a lot of subbasins. In our study Lavasanat, Damavand and Varamin are Salt Lake subbasins.
14. Line 192: The word “where” was added before explaining the variables and each variable was written in sentence.
15. Line 205: “(SPI)” was added.
16. Line 205: Typed error corrected (et al.)
17. Section 3.1: This section need complete rephrasing and expanding. It is very confusing and the emission scenarios results weren’t broken down in detail for explanation.
Response 17:
This section has been reviewed precisely and extensive description about SPI index, method of calculation, charts and climate scenarios have been added. In addition, section 2.3. has been added which specifically talks about climate scenarios.
18. Line 218: “Statistical analysis” was rephrased.
19. Line 236: “Sensitivity analysis” was rephrased.
20. Section 3.3 -3.5: These sections need expanding and much more elaborations. Need a much detailed interpretation of result and discussion. For instance, what are the driving forces and mechanisms to yield these results? Also, discuss and interpret each emission scenario result in much greater detail.
Response 20:
These sections have been reviewed and the paper has been revised. More details and descriptions regarding the results and climate scenarios have been added to the text.
21. “Because” was deleted.
22. Line 278: “a” replaced “A”.
23. Line 322: “Climate change” was rephrased.
24. Section 4: Again, what emission scenario is the conclusion base of? If it is all of them, you need to indicate here.
Response 24:
Results are based on all three climate scenarios (A1B, A2, and B1) that have been studied in this paper particularly A2 scenario. The results are valid for every single climate scenario. More details has been provided and got into description of the results based on climate scenarios.
All the comments have been addressed. All grammatical mistakes are highlighted in yellow and technical comments are highlighted in blue. Thank you for your time and consideration and I hope that I have addressed all the comments in a satisfactory manner.

Reviewer 2 Report
At first, I would like to congratulate the authors on the work developed. I have found interesting the study exposed here despite the fact that the main content does not show outstanding conclusions nor scientific soundness results. For this reason, I am going to expose next the most important points should be reviewed from my humble point of view:
Introduction: I think that the number of references to other works and studies is low. Most of the references mentioned not have continuity or reference in the results section. References should help to reinforce, validate or corroborate the results or the methodology.
Methods: the description of SPI index and the results obtained with this index should be highlighted due to is one of the key questions of the work. With this poor description, the reader can not discern the level of drought severity.
Calibration and validation dataset: The distribution of data among calibration and validation period should be revised: the common criterion is to divide into 2/3 and 1/3. In this case, the calibration period covers from 1998-2011 and the validation period only from 2011-2014
Results: I think that the author can get more information from the results developed. They have done a great job by the construction and development of the model but the outcomes are limited. Please, pay attention to the numbering of figures and tables, there is a general confusion over the whole manuscript.
Conclusions are very general, not reflect nothing really related to the work. It seems that can be applied to a typical paper of calibration-validation hydrologic model in any part of the world. IA greater implication of the results obtained is lacked
Author Response
Dear Second Referee:
Thank you for your time and recommendations. I have addressed your comments as follow:
1. Introduction: I think that the number of references to other works and studies is low. Most of the references mentioned not have continuity or reference in the results section. References should help to reinforce, validate or corroborate the results or the methodology.
Response 1:
Thank you for your invaluable comments. I have added more references to the text which agrees to the result of this paper. Since the majority of researches with SWAT model have focused on assessment of climate change on runoff, there are very limited studies that have focused on effects of climate change on aquifer percolation and renewable water resources. However I have added some new references to the text.
2. Methods: The description of SPI index and the results obtained with this index should be highlighted due to is one of the key questions of the work. With this poor description, the reader cannot discern the level of drought severity.
Response 2:
Section 3-1 which is related to SPI index has been reviewed and some additional information has been added to text. Section 2.3. Has been added to describe this part in more details. A new SPI index chart for a pried of (1986-2016) has been added which I believe helps to understand the concept better.
3. Calibration and Validation dataset: The distribution of data among calibration and validation period should be revised: the common criterion is to divide in to 2/3 and 1/3. In this case, the calibration period covers from 1998-2011 and the validation period only from 2011-2014.
Response 3:
In addition to laying the basis of 70% study period for calibration and 30% for validation, many research have been done based on 80% for calibration and 20% for validation.
4. Results: I think that the author can get more information from the results developed. They have done a great job by the construction and development of the model but the outcomes are limited. Please, pay attention to the numbering of figures and tables, there is a general confusion over the whole manuscript.
Response 4:
The results section has been reviewed and more details have been added. Numbering of the charts and tables have been revised as well.
5. Conclusions are very general, not reflect nothing really related to the work. It seems that can be applied to a typical paper of calibration-validation hydrologic model in any part of the world. IA greater implication of the results obtained is lacked.
Response 5:
More technical results have been added to results and discussion section. The result of each climate scenario has been discussed separately.
All the comments have been addressed. All grammatical mistakes are highlighted in yellow and technical comments are highlighted in blue. Thank you for your time and consideration and I hope that I have addressed all the comments in a satisfactory manner.

Reviewer 3 Report
Even if the theme is very interesting, the manuscript is lacking from the scientific point of view, the innovative aspect, that this manuscript should provide, is not evident. It is a simple application of SWAT to a case study.
Moreover, the text presents several punctuation, formatting and fonts problems for some figures.
My opinion is that the work is too weak to be published as it is
Author Response
Dear Third Referee:
Thank you for your time and recommendations. I have addressed your comments as follow:
1. Even if the theme is very interesting, the manuscript is lacking from the scientific point of view, the innovation aspect, that this manuscript should provide, is not evident. It is a simple.
Response 1:
This article has been reviewed and many details have been added to the text. In result and discussion part, technical results have been presented. More resources have been added to explain the main goal of this research. In addition from contribution aspect, I should note that there are very limited number of researches has been performed on climate change impact on renewable water resources under climate scenarios using SWAT model. This research not only estimated the impacts of climate change on surface water, groundwater, and renewable resources, but also drought intensity was predicted for future periods under climate scenarios.
2. Moreover, the text presents several punctuation, formatting and fonts problems for some figures.
Response 2:
All the numbering of figures and tables and font problems have been revised.
All the comments have been addressed. All grammatical mistakes are highlighted in yellow and technical comments are highlighted in blue. Thank you for your time and consideration and I hope that I have addressed all the comments in a satisfactory manner.

Round 2
Reviewer 1 Report
This version is fine.
Reviewer 2 Report
All questions proposed have been revised and corrected.
Reviewer 3 Report
The paper has been modified in such a way as to be considered now publishable